# Position: LLM Serving Needs Mathematical Optimization and Algorithmic Foundations, Not Just Heuristics

**Zijie Zhou** [1]

## Abstract

This position paper argues that LLM inference serving has outgrown generic heuristics and now demands mathematical optimization and algorithmic foundations. Despite rapid advances in serving systems such as vLLM and SGLang, their algorithmic cores remain largely unchanged from classical distributed computing: request routing uses join-shortest-queue or round-robin, scheduling defaults to FIFO, and KV cache eviction follows LRU. These general-purpose policies ignore the distinctive structure of LLM inference—dynamically growing KV cache memory, prefill-decode phase asymmetry, unknown output lengths, and continuous batching constraints. We contend that the field must develop mathematical models capturing these characteristics, enabling the design of algorithms with provable performance guarantees across diverse workloads, rather than heuristics that may succeed in some scenarios but fail unpredictably in others. Emerging work at the intersection of operations research and ML systems demonstrates that principled methods can match or exceed heuristic performance while providing theoretical guarantees. We call on the community to recognize algorithmic design for LLM serving as a research frontier.

## 1. Introduction

Large language models (Brown et al., 2020; Chowdhery et al., 2023; OpenAI, 2023; Kaplan et al., 2020) have evolved from research demonstrations into essential infrastructure powering search engines, coding assistants, and countless applications. The scale of deployment is immense: leading providers now serve billions of inference requests daily (Kim et al., 2025; Milmo, 2025; Roth, 2025), with energy consumption measured in gigawatt-hours (You, 2025). As demand continues its rapid growth, the efficiency of LLM serving has become both an economic imperative and an environmental necessity—at these scales, even modest improvements yield substantial reductions in cost and carbon footprint.

Realizing such improvements requires understanding what makes LLM inference structurally novel. Inference proceeds in two phases with fundamentally different profiles: a compute-bound *prefill* phase that processes the input prompt in parallel, and a memory-bandwidth-bound *decode* phase that generates tokens autoregressively. The KV cache—storing intermediate representations to avoid redundant computation—grows with each generated token, yet its final size remains unknown until generation completes. Continuous batching allows requests to enter and exit processing dynamically, coupling the fates of concurrent requests in complex ways. This combination of phase asymmetry, uncertain and growing memory consumption, and fluid batch composition creates a setting without direct precedent in classical distributed systems.

Recent years have seen substantial architectural innovations addressing these challenges. Continuous batching (Yu et al., 2022) allows requests to enter and exit batches dynamically rather than waiting for entire batches to complete, significantly improving GPU utilization. Paged attention (Kwon et al., 2023) manages KV cache memory in non-contiguous blocks, reducing fragmentation. Prefill-decode disaggregation (Zhong et al., 2024; Patel et al., 2024) assigns each phase to specialized worker pools matched to its computational characteristics. Mixture-of-experts (MoE) (Liu et al., 2024; DeepSeek, 2025b) architectures route tokens to specialized subnetworks, scaling model capacity without proportional compute cost. These advances, embodied in widely-adopted systems like vLLM and SGLang, have driven major gains in serving throughput and latency.

Yet beneath these architectural innovations, a gap remains. Each advance introduces decision problems: which requests to batch together, how to route requests across heterogeneous workers, when to evict cache entries, how to balance load across experts. How are these decisions made in practice? Largely through simple heuristics inherited

[1]Department of Industrial Engineering and Decision Analytics, HKUST. Correspondence to: Zijie Zhou <jerryzhou@ust.hk>.

*Proceedings of the 43rd International Conference on Machine Learning*, Seoul, South Korea. PMLR 306, 2026. Copyright 2026 by the author(s).

from classical distributed systems. Routing typically uses join-shortest-queue or round-robin. Scheduling defaults to first-in-first-out. Cache eviction follows least-recently-used. These general-purpose policies ignore the distinctive structure of LLM inference—the prefill-decode asymmetry, dynamic memory growth, uncertain output lengths, and the interdependencies introduced by continuous batching.

This paper argues that **LLM serving has outgrown generic heuristics and now demands rigorous mathematical optimization and algorithmic foundations.** The unique structure of LLM inference creates optimization problems that generic policies cannot exploit. Formal models of this structure enable algorithms with provable performance guarantees—ensuring reliability across diverse workloads, not just average benchmarks. Beyond empirical improvements, principled approaches yield theoretical insights: fundamental limits that guide capacity planning, and design principles that inform engineering practice. As serving architectures mature and stabilize around established paradigms, algorithmic innovations become durable investments—unlikely to require redesign with each incremental system update.

This opportunity is two-sided. On one hand, mathematical modeling and principled algorithm design can substantially advance LLM serving. The decision problems embedded throughout serving systems—request routing, scheduling, cache management, load balancing, capacity planning, resource allocation—are amenable to formal analysis. Emerging work demonstrates that approaches grounded in optimization theory can match or exceed heuristic performance while providing guarantees that heuristics cannot: worst-case bounds ensuring robustness under adverse workloads, fundamental limits informing capacity provisioning, and algorithmic structures that guide practical system design.

On the other hand, LLM serving poses genuinely novel problems for the optimization and algorithms community. The structural characteristics that make LLM inference distinctive—scheduling under dynamically growing memory constraints, caching items whose sizes expand during use, queues with service times revealed only mid-service—give rise to problem formulations without established solutions. These are not routine applications of classical techniques but new problem classes warranting dedicated study. The result is a productive exchange: serving systems gain efficiency and robustness guarantees, while the algorithms community gains a rich source of well-motivated, practically grounded challenges. This paper develops both sides of this opportunity.

## 2. The Algorithmic Landscape of LLM Serving

The LLM serving architectural innovations introduce decision problems that fundamentally shape system performance. This section examines several representative bottleneck problems, characterizing current practices and identifying opportunities for principled algorithmic approaches. We emphasize that this landscape is illustrative rather than exhaustive: algorithmic design opportunities permeate LLM serving, from batching and scheduling to cache management and capacity planning. For each problem, we describe its structure, the heuristics currently employed, and the gap that motivates formal optimization.

### 2.1. Expert Routing and Load Balancing in Expert Parallelism

Mixture-of-Experts (MoE) architectures (Shazeer et al., 2017; Fedus et al., 2022; Liu et al., 2024) scale model capacity by routing each token to a subset of specialized expert networks. In large-scale deployments, Expert Parallelism (EP) distributes these experts across multiple GPUs, with each device hosting a subset of experts. Tokens must be dispatched to remote GPUs via all-to-all communication, processed by the relevant experts, and gathered back—a communication pattern that repeats at every MoE layer. The central challenge is load imbalance: if tokens concentrate on a few popular experts, the GPUs hosting them become stragglers while others idle, and the overall latency is determined by the slowest device (see Figure 1).

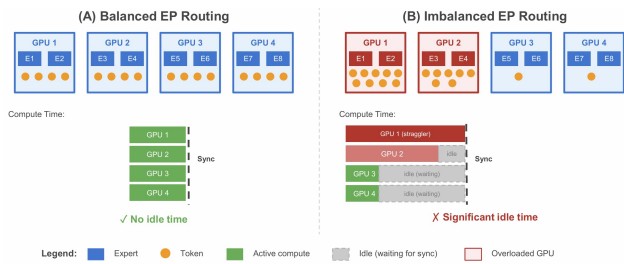

*Figure 1.* Under expert parallelism, imbalanced token routing causes straggler GPUs that delay synchronization. Lightly loaded GPUs must idle while waiting for heavily loaded GPUs to complete before all-to-all communication.

Current approaches to MoE load balancing rely primarily on heuristics developed during training. The most common strategy, introduced in GShard (Lepikhin et al., 2020) and refined in Switch Transformer (Fedus et al., 2022), adds an auxiliary loss that penalizes uneven token distribution across experts. While this encourages balance, it introduces interference gradients that conflict with the primary language modeling objective—practitioners must carefully tune the loss weight to trade off model quality against routing uniformity. Alternative heuristics include adding random noise

to routing scores, imposing expert capacity constraints that drop overflow tokens, and expert-choice routing where experts select tokens rather than vice versa (Zhou et al., 2022). More recently, auxiliary-loss-free methods dynamically adjust per-expert biases based on recent load (Liu et al., 2024; Wang et al., 2024), avoiding gradient interference but still relying on reactive heuristics without formal guarantees. Their effectiveness varies across workloads, batch sizes, and model scales.

The gap here is significant: expert routing during inference is fundamentally a constrained assignment problem—redistributing tokens across devices to minimize maximum load—yet it is addressed through indirect heuristics rather than direct optimization. This creates an opportunity for principled approaches. For instance, formulating load balancing as a linear program enables optimal token redistribution across redundant expert replicas in real time, achieving balance without auxiliary losses and their associated trade-offs. The result is not only improved performance but also greater interpretability and control: the optimization objective is explicit, constraints are transparent, and the solution is provably near optimal within the model. We elaborate on such approaches in Section 3.

## 2.2. Request Routing to Decode Workers and Load Balancing in Data Parallelism

Beyond expert-level routing, LLM serving systems must also route incoming requests to computational workers. In large-scale deployments using Data Parallelism (DP), multiple decode instances process requests in parallel, each maintaining its own KV cache and workload state. When these workers also employ Expert Parallelism internally, an additional constraint emerges: the EP communication (all-to-all dispatch and combine) requires synchronization across all workers. Before this synchronized phase, each worker processes its local attention computation, whose duration depends on the worker's current KV cache size. The fastest worker—with the lightest cache load—must wait for the slowest worker to complete before the collective EP phase can proceed (see Figure 2). Consequently, workload imbalance across decode workers directly translates to idle time and degraded throughput.

Current routing policy relies on general-purpose heuristics. The vLLM router (vLLM Project Contributors, 2025), for instance, offers five policies: round-robin, random selection, power-of-two-choices (selecting the less loaded of two randomly sampled workers), consistent hashing for session affinity, and cache-aware routing that considers prefix sharing. Similar strategies appear in SGLang (Zheng et al., 2024) and other serving frameworks. These policies originate from classical load balancing literature (Mitzenmacher, 2002; Karger et al., 1997) and apply broadly to parallel

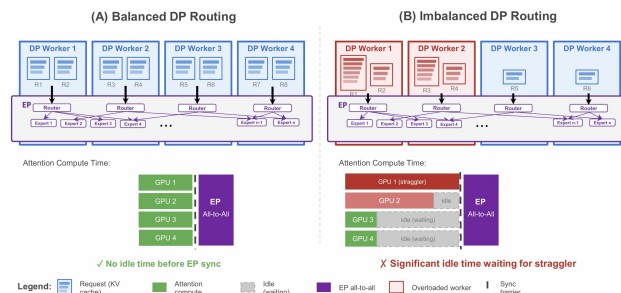

*Figure 2.* Under DP with internal EP, each worker computes attention locally before synchronizing for EP all-to-all communication. Workers with larger KV caches (longer sequences) take longer, forcing lightly loaded workers to idle at the sync barrier.

computing systems. However, they do not account for the specific structure of LLM decode workloads. First, the workload per request—determined by decode length—is unknown at routing time and reveals itself only as generation proceeds. Second, workloads drift in a predictable pattern: each active request's memory footprint grows by one KV unit per decode step, creating systematic load increases. Third, assignments are sticky: once a request is routed to a worker, migrating it would require transferring its entire KV cache, making re-routing prohibitively expensive. These characteristics—unknown duration, predictable growth, and irrevocable assignment—distinguish LLM routing from classical load balancing and suggest that tailored algorithmic approaches may yield substantial gains.

Request routing under barrier synchronization is also a constrained online assignment problem—minimizing the maximum load that gates progress at each step—yet current heuristics treat it as stateless dispatch. The problem's structure includes workload drift, sticky assignments, and unknown job durations, making it amenable to online integer programming formulations with provable worst-case guarantees (Chen et al., 2026). However, a direct optimization formulation requires quantities—such as decode lengths—that are unknown at decision time. This motivates a joint design perspective: identifying what minimal additional information enables tractable approximations, and developing algorithms that exploit precisely that information. We elaborate on such approaches in Section 3.

## 2.3. Scheduling and Capacity Planning within Individual Workers

Within each worker, continuous batching allows requests to enter and exit the batch dynamically at each decode step, rather than waiting for an entire batch to complete (Yu et al., 2022; Kwon et al., 2023). When a request finishes and a slot becomes available, the scheduler must decide which waiting request to admit. Current implementations typically default to first-come-first-serve (FCFS): vLLM appends ar-

riving requests to a waiting queue and admits them in arrival order (Kwon et al., 2023). While FCFS is simple, it ignores request characteristics—such as predicted output length or memory footprint—that affect both latency and throughput. Emerging work shows that alternative policies can improve performance: for instance, prioritizing shorter requests reduces average latency, while memory-aware admission prevents KV cache overflow. These are classical scheduling questions, now complicated by the LLM-specific structure of growing memory consumption and unknown job durations, suggesting opportunities for principled scheduling algorithms.

A related question is capacity planning: given a workload distribution, how many workers are required to keep the system stable—ensuring queues do not grow unboundedly? Current practice relies on reactive autoscaling heuristics, adjusting worker counts based on observed metrics such as queue depth, GPU utilization, or request concurrency (Stojkovic et al., 2025; Miao et al., 2024; Sun et al., 2024; Fu et al., 2024). However, scaling decisions incur latency—provisioning new workers takes time—during which an already-stressed system may deteriorate further. Queueing analysis offers a proactive alternative: by deriving closed-form stability conditions that account for both computation and KV cache memory constraints, operators can determine the minimum fleet size required for a given arrival rate before deployment, rather than discovering instability at runtime. Recent work demonstrates that such conditions can predict real-system behavior with high accuracy (Anonymous, 2025), illustrating how classical queueing theory, adapted to LLM-specific constraints, can inform practical capacity decisions. We elaborate on these approaches in Section 3.

### 2.4. Caching and Eviction Policies

Multimodal LLM serving—processing text alongside images, video, and audio—has become a critical workload as models evolve toward omni-modal capabilities (vLLM Community, 2025; Qiu et al., 2025; Liu et al., 2025; Ma et al., 2025). A key performance metric is time-to-first-token (TTFT), which in multimodal settings comprises image preprocessing, encoder inference, and language model prefill. Studies show that image encoding can dominate TTFT, contributing more than half of latency for vision-language models with high-resolution inputs (Qiu et al., 2025). A natural optimization is embedding caching: if an image or video has been previously encoded, its embeddings can be stored and reused, bypassing redundant encoder computation on subsequent requests (Red Hat Developer, 2025). However, embedding caches have limited capacity—GPU memory is constrained, and multimodal embeddings are large (a single high-resolution image may produce thousands of embedding vectors). When the cache is full and a new item arrives,

the system must decide which cached entry to evict. Current implementations, including vLLM's automatic prefix caching, default to least-recently-used (LRU) eviction: the entry accessed furthest in the past is removed first. While LRU is easy to implement, it ignores problem structure specific to LLM serving.

The caching problem in multimodal LLM serving differs from classical online caching and paging in several ways. First, cached objects have heterogeneous sizes: a short video clip produces far more embedding data than a thumbnail image, yet both compete for the same cache space. Second, the cost of a cache miss is content-dependent: re-encoding a high-resolution video is orders of magnitude more expensive than re-encoding a small image, so eviction decisions should account for recomputation cost, not just recency. Third, LLM workloads exhibit distinct temporal locality patterns—conversational sessions create bursty, correlated requests for the same media assets, while cross-session reuse may follow different popularity distributions. Recent work formalizes this setting and shows that combining cost-aware eviction policies with learned model selection can achieve optimal rates for minimizing total inference cost (Zhu et al., 2023). This suggests that principled algorithms can exploit the structure of LLM serving workloads to improve cache efficiency. We elaborate on these opportunities in Section 3.

## 3. The Opportunity: A Two-Sided Breakthrough

### 3.1. Why Theory Matters: Beyond Beautiful Mathematics

One might ask: if heuristics already work reasonably well in practice, why invest in formal optimization? The answer is that principled approaches provide four distinct benefits that heuristics cannot, each with direct practical implications.

**Worst-case guarantees ensure robustness.** Heuristics are typically validated on benchmark workloads that represent average or expected conditions. But production systems face adversarial and pathological scenarios: flash crowds during product launches, correlated failures during infrastructure incidents, and workload distributions that drift far from training data. A heuristic that performs well on ShareGPT traces may fail catastrophically when a viral application generates unusual request patterns. Algorithms with provable worst-case bounds—such as competitive ratios guaranteeing performance within a constant factor of optimal—provide insurance against such scenarios.

**Fundamental limits inform capacity planning.** Before deploying a serving cluster, operators must answer: how many GPUs are sufficient to handle a given request rate while meeting latency targets? Heuristic tuning cannot an-

swer this question—it can only reveal instability after the fact. Theoretical analysis, in contrast, can derive closed-form stability conditions that specify the minimum capacity required as a function of arrival rate, output length distribution, and memory constraints. These conditions serve as planning tools: they tell operators before deployment whether a proposed configuration will be stable, and if not, precisely what additional resources are needed. The alternative—reactive autoscaling that discovers instability at runtime—incurs both latency (provisioning takes time) and cost (over-provisioning for safety margins).

**Algorithmic structure guides engineering.** Even when a theoretically optimal algorithm cannot be deployed directly—perhaps it requires information unavailable at decision time, or its computational overhead is prohibitive—its structure often suggests practical approximations. A linear programming formulation, for instance, makes explicit which constraints are binding and which objectives matter. Engineers can then design heuristics that respect these constraints and approximate these objectives, rather than guessing at what factors might be important. The theory provides a blueprint; the engineering adapts it to operational constraints.

**Optimality baselines prevent over-engineering.** Without knowing how close current performance is to optimal, engineering effort can be wasted pursuing diminishing returns. Theoretical analysis provides baselines: lower bounds on achievable cost, upper bounds on achievable throughput, competitive ratios that benchmark online algorithms against offline optima. When a practical algorithm achieves performance within 5% of the theoretical limit, further optimization is unlikely to yield significant gains—effort is better directed elsewhere. When performance is 50% from optimal, substantial room for improvement exists. Theory calibrates expectations and allocates engineering attention efficiently.

### 3.2. A Historical Precedent: Theoretical Optimization Algorithms in Airline Revenue Management

The benefits articulated above are not hypothetical: they have already transformed industries through precisely the formulation-to-insight-to-deployment pipeline we advocate for LLM serving. The most illustrative precedent is airline revenue management, which shares striking structural similarities with LLM inference and demonstrates how mathematical optimization can deliver large-scale practical impact without sacrificing real-time efficiency.

Airlines face a canonical online resource allocation problem: seats are perishable capacity, booking requests arrive sequentially, each booking irrevocably consumes seat inventory, future demand is uncertain, and the goal is to maxi-

mize revenue subject to flight-leg capacity constraints. In the 1980s, American Airlines formulated this as a network linear program—maximizing total fare revenue subject to capacity constraints and demand forecasts (Smith et al., 1992). The LP's contribution, however, was not that airlines solve it for every incoming booking. Instead, the LP's *dual variables*—shadow prices on the capacity constraints—revealed the marginal value of each seat on each flight leg. This insight yielded an elegant *bid-price control* policy: accept a booking if and only if its fare exceeds the sum of bid prices on the flight legs it uses. The deployed policy is an $O(1)$ accept/reject rule requiring no solver at runtime, yet it is grounded in the LP's optimality conditions. American Airlines estimated $1.4 billion in incremental revenue over three years from this OR-based system, work recognized by the 1991 INFORMS Franz Edelman Award.

The structural parallel to LLM serving is direct. Seats on flights correspond to GPU memory and compute capacity; booking requests correspond to inference requests arriving online; irrevocable seat consumption corresponds to sticky KV cache assignment; unknown future demand corresponds to unknown output lengths; and the LP's bid-price insight corresponds directly to the optimization-derived thresholds and balancing policies we advocate for LLM routing and scheduling. Beyond airline revenue management, similar formulation-driven advances have transformed logistics and power systems, each grounded in mathematical programming yet deployed through fast, structurally-informed algorithms.

Crucially, this historical precedent illustrates exactly the scientific process our paper champions: (1) formulate the problem as a mathematical program; (2) analyze the formulation to extract structural insights—in this case, bid prices from LP duals; (3) deploy a fast, practical policy informed by these insights. The theoretical work, including the online LP framework of (Williamson, 1992) and (Agrawal et al., 2014), which generalizes bid-price methods with provable competitive ratios; the deployed systems inherited both the performance and the guarantees. The lesson is that mathematical optimization need not be deployed as a runtime solver to deliver enormous practical impact; its primary role is often as an *analytical vehicle* that reveals the structure of good algorithms. Section 3.3 demonstrates that this same pipeline is now beginning to take shape for LLM serving.

### 3.3. How Optimization Algorithms Can Improve LLM Serving

The problems introduced in Section 2 are not merely amenable to formal optimization—they have already begun to yield to it. We highlight three examples where principled approaches have demonstrated practical impact, illustrating both the techniques involved and the benefits they deliver.

**Example 1: Linear Programming for MoE Load Balancing.** For the expert routing problem described in Section 2.1, DeepSeek's LPLB system (DeepSeek, 2025b) demonstrates that per-batch token redistribution can be cast as an explicit optimization problem rather than addressed through tuned heuristics. LPLB extends EPLB (DeepSeek, 2025a): EPLB provides workload-driven expert reordering and selects which experts to replicate based on aggregate statistics, while LPLB addresses the dynamic, batch-to-batch fluctuations that remain even after static balancing.

LPLB represents the redundancy structure as a topology graph: each redundant expert induces an edge connecting the GPU hosting the original expert to the GPU hosting the replica. For a given batch, each edge has a capacity determined by that batch's token assignments—specifically, the number of tokens routed to the redundant expert. LPLB then solves a linear program to redistribute token workload along these capacity-constrained edges, e.g., minimizing the maximum per-GPU load within an expert-parallel group:

$$\min_{f, L_{\max}} \quad L_{\max}$$
$$\text{s.t.} \quad \ell_i - \sum_{j:(i\to j)\in E} f_{ij} + \sum_{j:(j\to i)\in E} f_{ji} \leq L_{\max} \quad \forall i,$$
$$\text{s.t.} \quad 0 \leq f_{ij} \leq c_{ij} \quad \forall (i \to j) \in E,$$

where $\ell_i$ is GPU $i$'s initial token load for the batch, $E$ is the set of directed edges induced by redundant experts, $f_{ij}$ is the token load shifted from GPU $i$ to $j$, and $c_{ij}$ is the edge capacity.

This LP is solved by an embedded GPU solver implementing the interior-point method (Luenberger & Ye, 2021). Importantly, the current implementation achieves optimality with respect to its stated objective (token-count balance) and constraints; it does not yet model the nonlinear cost structure of grouped matrix multiplications, which the authors note as a limitation. Nevertheless, the approach exemplifies how a scientific formulation can transform an indirect, heuristic-driven process into a direct, principled one: the optimization objective is explicit, the constraints are transparent, and within the model's scope, the solution is optimal by construction.

From a deployment perspective, LPLB also exemplifies a useful spectrum. EPLB, the optimization-informed heuristic, performs no runtime optimization yet is guided by the underlying assignment problem structure, and is currently deployed in DeepSeek V3/R1 production. LPLB takes the additional step of directly solving the LP per batch; its embedded GPU solver achieves approximately $100\mu s$ per solve for intra-node optimization, well within the typical $30$–$100ms$ decode-step budget. This timing demonstrates that direct optimization is operationally viable, not merely a theoretical aspiration.

**Example 2: Online Integer Optimization for DP Load Balancing.** For the request routing and DP load balancing problem described in Section 2.2, (Chen et al., 2026) develop an online optimization framework that addresses the barrier-synchronized, sticky-assignment setting of data-parallel LLM decoding. The problem structure differs fundamentally from the MoE load balancing of Example 1 in two respects. First, the objective is inherently nonlinear: under barrier synchronization, per-step idle time is exactly linear to the sum $\sum_{g=1}^{G}(L_{\max}(k) - L_g(k))$, where $L_{\max}(k) = \max_g L_g(k)$ is the maximum load across $G$ workers. Second, the problem is intrinsically online: requests arrive sequentially, each must be irrevocably assigned to a worker, and the state at decode step $k+1$ strongly depends on assignments made at step $k$ through the drift of KV growth. This temporal coupling means that greedy policies optimizing only the current step can accumulate errors that compound over time.

The key algorithmic insight is that effective load balancing does not require accurate prediction of total decode length—a notoriously difficult task—but only short-horizon forecasts of whether *active* jobs will complete within the next few steps. Leveraging this observation, the authors formulate a Balance-Future principle that, at each assignment step, solves an integer optimization minimizing accumulated predicted imbalance over a short lookahead window of $H$ steps. The decision variables are binary assignments of waiting requests to workers; the constraints encode capacity limits and the sticky-assignment rule; the objective sums the predicted imbalance $\sum_{h=0}^{H} \text{Imbalance}(k+h)$ using short-horizon workload forecasts for active requests. This formulation makes explicit that decode-time routing is not a stateless dispatch problem but a sequence of structured combinatorial optimizations with a barrier-driven objective.

A principal advantage of the optimization-based approach is that it admits rigorous worst-case analysis: the resulting guarantees hold regardless of the request distribution, dataset, arrival frequency, or ordering—properties that heuristics validated on specific traces cannot provide. Concretely, (Chen et al., 2026) prove that even when the arrival sequence is chosen adversarially, the algorithm reduces long-run average imbalance relative to the default policy by a factor of $\Omega(\sqrt{B \log G})$, where $B$ is the per-worker batch size and $G$ is the number of workers. This scaling is practically significant: the benefit of principled balancing *increases* with cluster size and batch capacity—precisely the regime in which large-scale serving systems operate. We refer readers to (Chen et al., 2026) for the detailed integer programming formulation and proofs.

**Example 3: Scheduling and Capacity Planning within Individual Workers.** Within each worker, the sophisticated process describes in Section 2.3 raises fundamental

questions: under what conditions is a worker stable, and which requests should be admitted when slots become available? A growing body of work applies queueing theory and combinatorial optimization to provide rigorous answers. On the stability front, (Anonymous, 2025) develops a queueing-theoretic framework that derives closed-form stability conditions accounting for both compute bound and memory bound—enabling operators to determine the minimum capacity required for a given arrival rate *before deployment*, rather than discovering instability at runtime. Building on this foundation, (Ao et al., 2025) formulate a multi-stage online scheduling problem and develop a fluid dynamics approximation that serves as a tractable performance benchmark. Their *Waiting for Accumulated Inference Threshold* algorithm uses threshold-based batching to keep the system near load balance, achieving near-optimal throughput under the stable system.

When the system is overloaded, the question shifts from stability to *which* requests to serve with priority. (Jaillet et al., 2025) formalize this as a combinatorial optimization problem that captures continuous batching and KV cache limits. A key contribution is the *hindsight optimal* benchmark: an integer program computing the minimum latency achievable by a clairvoyant scheduler with full knowledge of all arrivals and output lengths. By analyzing the LP relaxation's dual, they establish lower bounds that any online algorithm must respect—providing a principled baseline against which practical policies can be measured. When output lengths can be predicted, both (Jaillet et al., 2025) and (Shahout et al., 2024) show that shortest-job-first policies are near-optimal under some assumptions, as prioritizing short requests reduces waiting time and allows larger concurrent batches (since shorter requests consume less KV cache). These foundational models have spurred further algorithmic development: (Wang et al., 2025) extend the framework to long prefill lengths, developing novel batching criteria for long-prompt workloads; (Chen et al., 2025) address the practical challenge of inaccurate predictions by designing an adaptive algorithm that treats the predicted lower bound as the initial output estimate and dynamically refines it during inference, achieving a logarithmic competitive ratio even when predictions are highly uncertain.

**Example 4: Cost-Aware Caching with Optimal Regret.**
(Zhu et al., 2023) provides a principled optimization framework for the caching challenges described in Section 2.4 Their key insight is that eviction decisions should minimize *expected cost*, not just maximize hit rate: when re-encoding a high-resolution video costs orders of magnitude more than re-encoding a thumbnail, evicting the video to make room for the thumbnail is penny-wise but pound-foolish. To capture this, they design the Least Expected Cost algorithm, which scores each cached item by its cost-per-unit-size mul-

tiplied by its estimated access probability; when the cache is full, the item with the lowest score is evicted.

The theoretical contribution is achieving *optimal regret*. Regret quantifies this learning cost—the cumulative gap between the algorithm's decisions and those of a clairvoyant policy with perfect knowledge. Achieving optimal regret means the average per-query cost converges to the optimum as fast as information-theoretically possible; no algorithm can learn faster. (Zhu et al., 2023) prove that LEC combined with a learned model selector matches these lower bounds, providing the strongest possible guarantee for online cost minimization. Empirically, the combination yields up to $50\times$ cost reduction when the ratio between expensive and cheap operations is large, and achieves $4.3\times$ improvement in FLOPs and $1.8\times$ improvement in latency on real LLM workloads.

### 3.4. How LLM Serving Creates New Optimization and Algorithmic Problems

LLM inference exhibits two characteristics that distinguish it from classical computational workloads. First, the two-phase structure of inference—prefill followed by decode—creates a mixture of compute-bound and memory-bandwidth-bound operations within each request. Prefill processes the entire input prompt in parallel and is compute-intensive; decode generates tokens sequentially, repeatedly reading the KV cache, and is memory-bandwidth-bound. This heterogeneity means that optimal resource allocation differs between phases: prefill benefits from compute density while decode benefits from memory bandwidth, motivating disaggregated architectures that route each phase to specialized hardware. Second, memory consumption grows *during* execution: each decode step appends one KV entry per layer, so a request that generates $n$ tokens occupies memory proportional to its prompt length plus $n$. Unlike classical jobs whose resource footprint is fixed at arrival, LLM requests have dynamic, monotonically increasing memory demands that are unknown until generation completes.

These two characteristics—phase heterogeneity and growing memory footprint—transform familiar algorithmic problems into novel variants when applied to LLM serving. Classical scheduling (Brucker et al., 1998; Chen et al., 1998; Pinedo, 2012) assumes jobs have known or stochastically drawn service times; LLM scheduling must handle unknown decode lengths, phase-dependent resource profiles, and memory constraints that tighten as jobs progress. Classical bin packing (Yao, 1980; Coffman Jr et al., 1984) places items of fixed size into bins; LLM batching must pack items that *grow* after placement, with eviction as the penalty for overflow. Classical online resource allocation (Agrawal et al., 2014; Li et al., 2020) optimizes over many resource types with fixed per-job consumption; LLM re-

source allocation must jointly manage compute and memory bandwidth under the KV cache growth pattern, where the reward structure shifts between compute-bound and memory-bandwidth-bound regimes as requests progress through phases. Classical load balancing (Saeed et al., 2018; Rathi et al., 2024; Tarandeep & Bhushan, 2020) distributes jobs across servers to equalize utilization; LLM load balancing under barrier synchronization must equalize the *maximum* load, with workloads drifting deterministically over time. Classical caching (Sleator & Tarjan, 1985; Fiat et al., 1991; Lykouris & Vassilvitskii, 2021) evicts items based on recency or frequency; LLM caching for multimodal embeddings must account for heterogeneous object sizes and content-dependent miss costs. Queueing theory also requires extension (Mitzenmacher & Shahout, 2025): the standard notion of offered load must incorporate memory as a first-class constraint, since a system can be compute-stable yet memory-unstable. In each case, the LLM-specific structure creates problem variants that existing theory does not directly address, opening new directions for algorithmic foundations research.

## 4. Alternative Views

The case for principled algorithmic approaches to LLM serving is not without skepticism. We list four common counterarguments, acknowledging their validity while explaining why they do not diminish the value of optimization foundations.

**View 1: Heuristics are good enough in practice.** Production systems like vLLM and SGLang serve millions of requests daily using relatively simple policies. If these heuristics work at scale, why invest in more sophisticated approaches? We acknowledge that heuristics have enabled remarkable scale and that "good enough" is often the pragmatic engineering choice. However, "good enough" obscures real costs. For example, unpredictable performance under workload shifts, and idle resources during load imbalance all represent inefficiencies that compound at scale. Given the substantial infrastructure investments in large-scale LLM deployment, even modest improvements in throughput or energy efficiency can yield significant operational savings. Moreover, heuristics validated on current workloads may fail silently on future workloads—a risk that principled approaches with formal guarantees can mitigate.

**View 2: The problem evolves too fast for theory to keep up.** Hardware generations turn over rapidly; model architectures shift from dense transformers to mixture-of-experts to state-space models; workload distributions change as applications evolve from chatbots to agents to long-context reasoning. By the time a theoretical result is published, the system it models may be obsolete. We acknowledge this

tension—LLM serving is a fast-moving target, and structural changes like the shift to decode-only transformers have fundamentally altered the problem landscape. However, even when specific problem structures evolve, the insights derived from theoretical analysis often remain valuable. Understanding why barrier synchronization creates stragglers, why unknown job sizes complicate scheduling, or why memory growth patterns induce instability provides conceptual clarity that transfers across system generations. These insights inform the design of future heuristics and help engineers anticipate failure modes before they manifest in production. In contrast, a heuristic tuned purely through empirical iteration on one system offers no such transferable understanding—it must be re-engineered from scratch when the underlying system changes.

**View 3: Systems engineering dominates algorithmic gains.** Kernel fusion, memory pooling, FlashAttention, speculative decoding, and hardware-aware compilation have delivered order-of-magnitude speedups. Compared to these systems-level optimizations, algorithmic improvements in scheduling or routing may seem marginal. We agree that systems engineering is critical and that no scheduling algorithm can compensate for an inefficient attention kernel. But this framing presents a false dichotomy. Algorithmic and systems improvements are complementary, not competing. A well-designed scheduler amplifies the gains from a fast kernel by keeping the GPU saturated; a poorly designed one squanders them through load imbalance or memory thrashing. Moreover, algorithmic analysis can *guide* systems design: knowing the stability region of a queueing system informs hardware provisioning; understanding the structure of optimal batching policies informs memory allocator design. The most efficient systems will combine both.

**View 4: Provable guarantees don't translate to real-world gains.** Worst-case competitive ratios are often loose; a 2-competitive algorithm may perform identically to an optimal one on typical inputs, making the guarantee seemingly vacuous in practice. Conversely, an algorithm with poor worst-case bounds may excel on real workloads. We acknowledge that worst-case analysis has limitations and that empirical performance is the ultimate metric. However, the value of theoretical guarantees lies not in claiming superiority over heuristics, but in establishing *universality*: a competitive ratio ensures that performance holds across any data distribution, arrival pattern, or adversarial input—not just the benchmarks on which a heuristic was tuned. This universality provides a foundation that practitioners can build upon. Once the structure of a provably good algorithm is understood, engineers can adapt it into practical heuristics that preserve the core insights while accommodating system-specific constraints. The examples in Section 3 il-

lustrate this synergy: principled formulations reveal which constraints bind and which objectives matter, guiding the design of efficient approximations that inherit robustness from their theoretical foundations. Theory and heuristics are not adversaries; theory provides the scaffolding on which better heuristics can be constructed.

# 5. Conclusion and Future Research Directions

This paper has argued that LLM inference serving has reached a stage where principled optimization and algorithmic foundations can deliver substantial benefits over ad-hoc heuristics. The unique structure of LLM serving creates optimization problems that generic policies cannot exploit. Emerging work demonstrates that formal approaches can match or exceed heuristic performance while providing guarantees that heuristics cannot: worst-case robustness, fundamental limits for capacity planning, and algorithmic structures that guide practical engineering.

We highlight several concrete research directions that we believe are ripe for progress:

**Scheduling with predictions under uncertainty.** Current work assumes predictions of varying quality, but the joint design of prediction models and scheduling algorithms remains underexplored. How should algorithms adapt when prediction quality varies across request types or drifts over time? What are the optimal robustness-consistency tradeoffs when predictions may be adversarially wrong?

**Multi-objective optimization.** Production systems must balance multiple metrics simultaneously: time-to-first-token, time-per-output-token, throughput, energy consumption, and fairness across users. Formalizing these as multi-objective optimization problems and characterizing Pareto frontiers would provide principled guidance for system operators navigating these tradeoffs.

**Theoretical foundations for disaggregation.** Prefill-decode disaggregation has emerged as a practical architecture, but its theoretical underpinnings remain unclear. Under what conditions does disaggregation outperform co-location? What is the optimal resource allocation between prefill and decode pools as workload characteristics vary?

**Algorithmic foundations for agentic inference.** As LLMs increasingly operate as agents—making tool calls, engaging in multi-turn reasoning, and spawning sub-requests—new scheduling challenges emerge. These workloads exhibit branching, variable-length pauses, and dependencies that existing models do not capture. Formalizing and solving scheduling problems for agentic workloads is an emerging frontier.

We call on researchers from both communities to engage with these challenges through genuine cross-disciplinary learning. For optimization and algorithms researchers, we encourage deep engagement with the systems literature—not just abstracting away implementation details, but understanding which bottlenecks dominate in practice, which constraints are hard versus soft, and which simplifying assumptions break down at scale. Models that fail to capture real system behavior may yield elegant theory but limited practical impact. For systems researchers, we encourage reading the optimization and algorithms literature to understand the structural insights that principled analysis can reveal: what fundamental limits exist, and how good algorithms are constructed. These insights can inform heuristic design even when the full theoretical algorithm is impractical to deploy. The intersection of these communities offers the potential for advances that neither could achieve alone.

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
