# OpenReview forum: "Position: LLM Serving Needs Mathematical Optimization and Algorithmic Foundations, Not Just Heuristics"
_ICML.cc/2026/Position_Paper_Track — ICML 2026 Position Paper Track regular_

### Official Review · Reviewer_2G1g · 2026-03-09

**Significance:** 2
**Argument Clarity:** 2
**Rating:** 4
**Confidence:** 4

**Questions:**

1. Is the underlying optimization problem NP-hard in general? If so, how do the authors reason about feasibility and approximation guarantees in practice?
2. How does the proposed global optimization framework adapt to dynamic, and highly skewed workloads that are common in real-world LLM deployments?

**Alternative Views Section:**

Yes

**Compliance With Llm Reviewing Policy A Conservative:**

Affirmed.

**Discussion Potential:**

2

**Final Justification:**

The rebuttal addressed some weaknesses.

**Paper Summary:**

This paper argues that LLM serving systems should be grounded in principled mathematical optimization and algorithmic foundations, rather than relying on ad hoc heuristics that may fail to generalize across workloads or deployment settings.

**Position:**

Yes

**Position In Title:**

Yes

**Related Work:**

2

**Strengths And Weaknesses:**

Strengths:

The paper advocates for a mathematically grounded optimization framework, which has the potential to provide better guarantees than heuristic-based designs, particularly in worst-case scenarios.

Weaknesses / Concerns:

1. It is unclear whether the proposed global optimization formulation is computationally tractable. The paper does not clearly characterize whether the underlying problem is NP-hard.
2. The paper lacks a clear explanation of how the global optimization approach adapts to highly dynamic and skewed workloads, which are common in real-world LLM serving environments.
3. It remains unclear how the proposed optimization framework accommodates different hardware characteristics.

**Support:**

2

---

> ### Author Rebuttal · Authors · 2026-03-28
>
> We thank Reviewer 2G1g for the review. We address each concern below with concrete details.
>
> **On tractability and NP-hardness.**
>
> We first clarify that the paper does not propose a single ``global optimization formulation.'' It presents multiple *distinct problems*, each with different complexity:
>
> -**MoE load balancing (Example~1):** A linear program---polynomial-time solvable. DeepSeek's LPLB solves it in ${\sim}100\mu s$ on a single GPU SM.
>
> -**DP routing (Example~2):** An integer program---NP-hard in general. However, the short lookahead horizon $H$ and small number of binary variables per step make each instance tractable in practice under approximation algorithms.
>
> -**Capacity planning (Example~3):** Closed-form analytical conditions derived from queueing theory---no optimization problem to ``solve'' at runtime.
>
> -**Caching (Example~4):** An online algorithm with provable optimal regret---$O(1)$ per decision.
>
> This diversity is precisely the point: *formalizing* the problems reveals their complexity class, which determines the right algorithmic approach. Some admit exact polynomial solutions; others require approximation algorithms with provable guarantees. Without formalization, one cannot even know whether a problem is easy or hard---or how close a heuristic is to optimal.
>
> We emphasize that our position is *not* that optimization should replace heuristics outright (see our responses to Reviewers xynm and tKwY for a detailed discussion). We advocate a scientific methodology: formulate the problem, analyze its structure, then design practical algorithms---possibly fast approximations---that inherit theoretical guarantees. Even for NP-hard problems, this process yields algorithms with provable approximation ratios, bounding how far any practical solution can deviate from optimal. Designing heuristics without this foundation means operating blind to both the problem's difficulty and the heuristic's quality.
>
> **On dynamic and skewed workloads.**
>
> We respectfully suggest that dynamic and skewed workloads are precisely where optimization-based approaches offer the *largest* advantage over heuristics, not the smallest.
>
> Consider an analogy: GPS navigation vs the heuristic ``always take the highway.'' Under normal traffic, the heuristic works well. But when an accident causes skewed congestion, the heuristic keeps routing cars onto the blocked road, while GPS---using a mathematical shortest-path optimization whose parameters (edge travel times) update with conditions---adapts immediately. The optimization framework is unchanged; only the inputs change. Skewed, dynamic conditions are exactly when the model is needed most.
>
> Concretely, the DP routing algorithm (Example 2) provides worst-case guarantees that hold for any arrival sequence, including adversarially skewed ones. The $\Omega(\sqrt{B\log G})$ improvement factor is distribution-free---it does not assume benign or balanced workloads. Round-robin, by contrast, performs well when requests are homogeneous but fails when skewed workloads cause long requests to cluster on one worker, creating stragglers at synchronization barriers. The more skewed and dynamic the workload, the larger the gap between optimization-based and heuristic approaches.
>
> **On hardware heterogeneity.**
>
> Hardware heterogeneity is a *strength* of the optimization framework, not a limitation. In a mathematical formulation, hardware differences are simply *parameters*: worker $g$ has memory capacity $M_g$ and compute throughput $s_g$, which appear as constraint coefficients. Swap A100s for H100s? Change the parameter values; the framework is unchanged. The optimization naturally routes memory-intensive requests to high-memory workers and compute-heavy prefills to faster GPUs, because these trade-offs are encoded in the objective and constraints.
>
> Compare this with round-robin: it treats all workers identically. To handle heterogeneous hardware, one must patch it into ``weighted round-robin''---but what weights? The right weight depends on the *current* workload mix (ratio of long vs short requests, prefill vs decode load), which is dynamic. Setting weights manually amounts to re-deriving an optimization problem through ad-hoc tuning. The formulation makes this principled from the start: heterogeneity adds constraints, and the solver (or the approximation algorithm derived from the formulation) respects them automatically.
>
> **On support, significance, and discussion potential.**
>
> We note that all qualitative dimensions were rated ``fair'' without specific elaboration. We hope the concrete clarifications above---particularly the complexity characterization, the distribution-free guarantees under skewed workloads, and the natural handling of hardware heterogeneity---address the underlying concerns. We welcome further discussion on any aspects the reviewer found lacking.

---

> > ### Author Rebuttal · Reviewer_2G1g · 2026-04-07
> >
> > Thank you for the authors’ responses. I have no follow-up questions.

---

### Official Review · Reviewer_Tros · 2026-03-11

**Significance:** 3
**Argument Clarity:** 2
**Rating:** 4
**Confidence:** 3

**Questions:**

Could the authors comment on the arguments about the solver time? For example, for the DP routing program, what is the wall-clock time relative to decode steps, and how does this scale with cluster scale?

Could the authors elaborate on the empirical substance of the provided bounds that theoretical optimization programs provide? The paper argues for the real-world use; thus, this would benefit from real-world numbers.

How sensitive are the queueing models to misspecification, such as correlated arrivals and heavy-tailedness, when applied to capacity planning?

**Alternative Views Section:**

Yes

**Compliance With Llm Reviewing Policy A Conservative:**

Affirmed.

**Discussion Potential:**

3

**Final Justification:**

The rebuttal clarified some weaknesses, warranting a rise in the score.

**Paper Summary:**

This paper argues that LLM inference serving should transition from heuristics to mathematical optimization and algorithmic foundations. The paper mainly addresses four different bottlenecks: MoE routing, data-parallel routing, within-worker scheduling and caching. The authors put forward that each of these can be addressed by heuristics that ignore LLM-specifics, but could be improved with optimization-based approaches such as linear programming, online integer programming, queue planning, and cost-aware caching.  Further, the authors argue that these methods provide benefits that heuristics cannot, such as optimality and worst-case bounds.  Finally, the paper proposes that LLM-serving creates novel optimization problems for the research community.

**Position:**

Yes

**Position In Title:**

Yes

**Related Work:**

3

**Strengths And Weaknesses:**

Strengths:

- [Choice of problematic areas] The four chosen problematic areas have a significant impact and are well motivated. The sophisticated LLM architectures, but the heuristics employed to infer them are a well-chosen mismatch and clearly described.
- [Presentation] The paper is well written and accessible to researchers as a call to action.

Weaknesses:

- [Limited mathematical foundation] For a paper calling for mathematical foundations, the mathematical foundation is limited. The mathematical LPLB formulation is optimal only within a model that omits important real-world complexities (nonlinearity), and it is unclear that it would beat a heuristic in terms of cost and performance (as noted in the original paper), countering the main argument of the authors.

- [No overhead analysis] Solver overhead and computational overhead of the optimization problems the paper asks for remain unaddressed. This can be an important reason for such formulations not to be employed and researched.  At run-time, the formulations are often relaxed for efficiency, and it is unclear if they then remain relevant. The paper would be improved with an analysis of how the latency of such solvers impacts inference serving.

- [Unclear capacity planning argument] The capacity planning argument is not well-defined. The queueing theoretic conditions rely on specific knowledge and parametrizations of arrival rate and the length distribution, but the authors note the system is unpredictable in their argument against heuristics. Model misspecification could render such a theory irrelevant.

**Support:**

2

---

> ### Author Rebuttal · Authors · 2026-03-28
>
> We thank Reviewer Tros for the detailed review. We address each concern below.
>
> **On limited mathematical foundation.**
>
> We respectfully believe this reflects a misreading of our argument. The reviewer states that LPLB ``is unclear that it would beat a heuristic,'' suggesting this counters our position. However, our position is *not* that any single formulation is a deployable solution dominating all heuristics. We argue that the **formulation-based methodology**---making objectives explicit, constraints transparent, and solutions provably optimal within the model---is the right approach for advancing LLM serving.
>
> LPLB illustrates this. The LP omits nonlinear grouped-GEMM costs, as we acknowledge. But its value was *analytical*: it identified the right objective (minimize max per-GPU load), revealed exploitable structure (the redundancy topology graph), and guided EPLB---the optimization-informed heuristic now deployed in DeepSeek V3/R1 production. Incorporating nonlinear costs is a natural extension the formulation *enables*---e.g., extending to a convex program or using the LP as a warm start. Without the initial formulation, such extensions have no foundation.
>
> For a *position paper*, the standard is to argue a research direction deserves study---not to solve all problems. Our four worked examples with formal formulations and provable guarantees exceed the mathematical depth typical of position papers, and each demonstrates the methodology we advocate.
>
> **On solver overhead.**
>
> We address this in detail in our responses to Reviewers xynm and tKwY: LPLB's GPU solver runs in ${\sim}100\mu s$ (decode steps take ${\sim}30$--$100ms$); the DP routing IP uses a short lookahead with few binary variables per step; and when direct solving is too slow, formulations yield $O(1)$ runtime policies via structural analysis (see the airline revenue management parallel in our response to Reviewer tKwY). We will add this discussion to the revision.
>
> On whether relaxed formulations ``remain relevant'': LP relaxations provide dual bounds benchmarking any heuristic's quality, and their solutions reveal binding constraints guiding algorithm design. Relaxations are analytical tools, not concessions---this is standard OR methodology.
>
> **On capacity planning and model misspecification.**
>
> The reviewer identifies a genuine tension: we argue systems are unpredictable, yet queueing models require distributional assumptions. We address this directly.
>
> This is not a contradiction. Heuristics fail because they model no structure---a threshold like ``scale up when utilization $> 80\%$'' has no analytical relationship to the workload. Queueing theory models the dominant structure (compute/memory constraints, arrival rates) even with approximate inputs. A 20\% error in arrival rate yields a proportional capacity error---far more informative than a reactive heuristic discovering instability only after SLO violations.
>
> Moreover, queueing results provide *scaling relationships and necessary conditions*, not point predictions. ``You need at least $m \geq \lceil \lambda / \mu \rceil$ GPUs'' remains useful even when $\lambda$ is imprecise, because it establishes the structure of the capacity-workload relationship. This is analogous to structural engineering: simplified beam models guide bridge design despite uncertain loads, because approximate analysis dominates no analysis.
>
> The framework is also *complementary to* reactive autoscaling. Queueing analysis provides the steady-state operating point that autoscaling ultimately converges to---but reaches it at deployment time, avoiding the cost of reactive discovery (GPU provisioning and KV cache warm-up take minutes). Practically, an autoscaler can use closed-form capacity estimates with updated arrival rates to scale proactively.
>
> Regarding sensitivity to correlated arrivals and heavy tails: this is an excellent research question that the position paper's framework enables---robust queueing under distributional uncertainty is a natural extension once the baseline model exists, and precisely the investigation our call to action aims to stimulate.
>
> **On empirical substance of bounds.**
>
> Numbers from cited works: the DP routing algorithm achieves $\Omega(\sqrt{B\log G})$ imbalance improvement over defaults; cost-aware caching achieves up to $50\times$ cost reduction and $1.8\times$ latency improvement on real LLM workloads. Comprehensive benchmarking across diverse configurations is the domain of dedicated technical papers; the position paper's role is to argue these formulations deserve such study.

---

> > ### Author Rebuttal · Reviewer_Tros · 2026-04-04
> >
> > Thanks for the detailed rebuttal. I have no further questions. I will adjust the current rating.

---

### Official Review · Reviewer_tKwY · 2026-03-13

**Significance:** 3
**Argument Clarity:** 4
**Rating:** 5
**Confidence:** 5

**Questions:**

I'd appreciate if the authors could discuss my weaknesses in detail.  I'd be happy to discuss throughout the discussion period as well.

**Alternative Views Section:**

Yes

**Compliance With Llm Reviewing Policy A Conservative:**

Affirmed.

**Discussion Potential:**

3

**Final Justification:**

See comment in rebuttal for more detail.  Overall, with the additional motivation, computational context, and limitations discussed in more detail, this would improve the submission.

**Paper Summary:**

This paper argues for the deeper integration of mathematical optimization, primarily from the operations research (OR) perspective, to improve LLM Serving.  The authors argue a reasonable case for researchers in the machine learning (ML) and OR communities to focus on how more principled algorithms, e.g., those with theoretical guarantees, for LLM serving provide benefits.  In addition, the authors highlight a few key tasks, e.g., MoE Load Balancing, that are formulated as constrained optimization problems.   They also argue fair critiques of the position, such as the acknowledgement of the fast-evolving ecosystem for LLM serving make this challenging, and that heuristics may be good enough in some cases.

**Position:**

Yes

**Position In Title:**

Yes

**Related Work:**

3

**Strengths And Weaknesses:**

### Strengths
-   **Position**: The authors make clear arguments both for and against their position, with well-supported reasons.  Given the complexity of serving LLMs and the potential improvements that mathematical optimization offers to the ML community, I'd say the position is reasonable and effectively articulated and argued.
- **Examples**: The authors provide clear examples of common LLM serving problems and their formulation as optimization problems.  This is a great inclusion that provides clarity on the problem's significance and intuition, especially for those less familiar with OR.

### Weaknesses:
- **Complexity**:  One aspect not fully discussed, which may be a major limitation for the application of optimization for LLM serving, is the complexity of actually solving the problems at scale.  While mathematical optimization offers principled algorithms with guarantees, the ability to deploy them at scale is a major limitation that is not discussed.  For example, the authors discuss multiobjective optimization and optimization under uncertainty.  However, the scalability of these algorithmic paradigms, in particular, is likely far from being usable in the context of large-scale allocation-type problems, especially those with discrete decisions.  A discussion of where these scalability gaps, e.g., the problem sizes current algorithms can address, and the current size of these LLM serving problems in practice, would be useful as an *Alternative View*.  Moreover, this might more concretely narrow high-impact application areas.
- **Real-time decision-making**:  Generally, any mathematical optimization algorithm will be much slower than heuristics, so discussing the complexity of optimization methods to compete in real-time large-scale environments would be useful to include.
- **Background**: One area of related work that would be useful to include is a stronger background on where exactly mathematical optimization has been very valuable in decision-making, e.g., power systems, logistics, etc. Given that a majority of the ML community is not particularly familiar with OR, providing a strong motivation for where OR has been successful in the past would provide better context.  Moreover, applications such as power systems are directly related to power delivery, which has undergone a significant shift in the era of data centers, and would be good to discuss given they're already of significant focus, heavily rely on mathematical optimization, and are related to LLM serving (albeit from a slightly different perspective than the paper focuses on).

**Support:**

3

---

> ### Author Rebuttal · Authors · 2026-03-28
>
> We sincerely thank Reviewer tKwY for the thorough and constructive review, the recognition of our position's clarity and examples, and the explicit invitation to discuss during the rebuttal period. We address each weakness below.
>
> **On complexity/scalability (Weakness 1) and real-time decision-making (Weakness 2).**
>
> We address these two points together, as they share a common concern: whether principled optimization can operate within the time and scale constraints of production LLM serving. We agree this is an important question that deserves more discussion, and we will expand on it in the revision. We have discussed thoroughly in the second point of the rebuttal of Reviewer xynm, and here is the summarization:
>
> The paper's core claim is *not* that one should solve large-scale optimization at every decision point; rather, optimization formulations yield structural insights guiding fast, practical algorithms (see our response to Reviewer xynm for this methodology). For problems where direct solving is feasible: DeepSeek's LPLB solves the MoE load balancing LP in ${\sim}100\mu s$ on a single GPU SM, while a typical decode step takes ${\sim}30$--$100ms$---making real-time LP solving practical within the existing time budget. For the DP routing problem (Example 2), the integer program operates over a short lookahead horizon $H$ with binary assignment variables per waiting request, keeping the per-step problem small even as the cluster scales.
>
> We acknowledge the reviewer's valid point that multiobjective optimization and optimization under uncertainty are harder to scale, particularly for large discrete problems. We will clarify in the revision that these are identified as research directions (Section 5), not currently deployable paradigms. This distinction helps narrow the high-impact areas as the reviewer suggests: near-term impact is strongest for problems with tractable structure (LP-based MoE balancing, short-horizon online assignment, queueing-theoretic capacity planning), while multiobjective and robust formulations represent longer-term opportunities requiring algorithmic advances.
>
> **On OR success stories as motivation (Weakness 3).**
>
> We appreciate this suggestion and will add a discussion of OR's real-world impact in the revision. We highlight one example that is not only historically significant but also structurally parallel to LLM serving: airline revenue management.
>
> Airlines face a canonical online resource allocation problem: seats are perishable capacity, booking requests arrive sequentially, each booking irrevocably consumes seat inventory, future demand is uncertain, and the goal is to maximize revenue subject to capacity constraints. In the 1980s, American Airlines formulated this as a *network linear program*---maximizing total fare revenue subject to flight-leg capacity constraints and demand forecasts (Smith, Leimkuhler & Darrow, 1992). The LP's contribution was not that airlines solve it for every incoming booking. Instead, the LP's dual variables (shadow prices) revealed the marginal value of each seat on each flight leg, yielding an elegant bid-price control policy: accept a booking if its fare exceeds the sum of bid prices on the flight legs it uses. This is an O(1) accept/reject rule, requiring no solver at runtime, yet it is grounded in the LP's optimality conditions. American Airlines estimated $1.4 billion in incremental revenue over three years from this OR-based system, winning the 1991 INFORMS Franz Edelman Award.
>
> The structural parallel to LLM serving is direct. Seats on flights correspond to GPU memory/compute capacity; booking requests correspond to inference requests arriving online; irrevocable seat consumption corresponds to sticky KV cache assignment; unknown future demand corresponds to unknown output lengths; and the LP's bid-price insight---that capacity has a marginal value guiding accept/reject decisions---corresponds to the optimization-derived thresholds and balancing policies we advocate for LLM routing and scheduling.
>
> Crucially, the airline revenue management story illustrates exactly the scientific process our paper champions: (1) formulate the problem as a mathematical program; (2) analyze the formulation to extract structural insights (bid prices from LP duals); (3) deploy a fast, practical policy informed by these insights (bid-price control). The theoretical work (including the Online LP framework of Agrawal et al., 2014, already cited in our paper, which generalizes bid-price methods with provable competitive ratios) came first; the deployed systems inherited both the performance and the guarantees.
>
> This historical precedent is compelling: airline revenue management transformed the industry through the same formulation-to-insight-to-deployment pipeline we advocate, applied to a problem with remarkably similar structure. We will include this discussion, along with additional OR successes in logistics and power systems, in the revision.

---

> > ### Author Rebuttal · Reviewer_tKwY · 2026-04-04
> >
> > See above comment

---

### Official Review · Reviewer_xynm · 2026-03-13

**Significance:** 3
**Argument Clarity:** 3
**Rating:** 4
**Confidence:** 4

**Questions:**

How should practitioners decide when a principled optimization method is deployable versus when it should only serve as a design guide for heuristics? The paper argues well for theory as scaffolding, but a clearer criterion on whether to deploy based on runtime overhead, observability, or robustness would further benefit practitioners.

**Alternative Views Section:**

Yes

**Compliance With Llm Reviewing Policy A Conservative:**

Affirmed.

**Discussion Potential:**

3

**Final Justification:**

With the proposed revision plan in rebuttal, this paper can be further strengthened. I'll maintain my original positive score.

**Paper Summary:**

This position paper argues that LLM serving now needs mathematical optimization and algorithmic foundations tailored to its structure, rather than relying on heuristic policies such as round-robin routing, FIFO scheduling, and LRU-style eviction. The core motivation is that LLM inference has distinctive properties, i.e. prefill-decode asymmetry, continuously growing KV-cache memory, unknown output lengths, and continuous batching, which are not well captured by classical distributed-systems heuristics.

**Position:**

Yes

**Position In Title:**

Yes

**Related Work:**

3

**Strengths And Weaknesses:**

Strengths:

1. Clear and timely position: the position is easy to identify and consistently argued in the paper: LLM serving has structural properties that deserve tailored algorithmic treatment rather than heuristics.
2. Good problem framing: the paper translated serving practice into recognizable optimization problems, especially for MoE balancing, worker assignment, and cache eviction under heterogeneous costs.


Weakness:

1. Many cited supporting works appear early-stage. Several references are recent arXiv or under submission, which is understandable for a fast-moving area, but it weakens the maturity of the evidence for a strong claim.

**Support:**

3

---

> ### Author Rebuttal · Authors · 2026-03-28
>
> We sincerely thank Reviewer xynm for the thoughtful review and the recognition of our paper's position, problem framing, and consistency of argument.
>
> **On early-stage references.**
>
> We acknowledge that several cited works are recent preprints. We offer an additional perspective: a position paper is most valuable when it addresses an *early-stage* topic and calls for broader community engagement. If the algorithmic foundations for LLM serving were already mature with textbook solutions, a position paper would be unnecessary---the case would be common sense. That many works are recent is evidence the field is at the right stage for a call to action: early enough for community investment to shape the agenda, yet developed enough that concrete examples (Section 3.2) demonstrate feasibility. We also note that while the *applications* are new, the underlying techniques---linear programming, competitive analysis, queueing theory---draw on decades of established Operations Research (OR) foundations.
>
> **On when to deploy optimization and use it as a design guide.**
>
> We thank the reviewer for this excellent question, which touches the heart of our message.
>
> *Our core claim is not that theoretical algorithms should replace heuristics, but that they **complement each other** through a scientific process*: (1)observe the problem structure; (2)formulate a mathematical optimization model capturing key trade-offs; (3)analyze the formulation for structural insights, bounds, and optimality properties; (4)use these insights to design practical approximation algorithms that possibly inherit theoretical guarantees. This produces far better algorithms than designing heuristics from intuition alone, because the formulation reveals *which constraints bind and which objectives matter*.
>
> This is well-established in OR. A classical and related example is Online Linear Programming (Agrawal et al., 2014, cited in the paper), which formulates sequential allocation as an LP at each period. Its contribution is **not** solving these LPs every step---instead, LP structural analysis yields a lightweight *bid-price control* policy requiring no solver at runtime, yet achieving a provable near-optimal performance against the hindsight optimum. The LP was the analytical vehicle; the deployed algorithm is fast and practical.
>
> We see this pattern emerging in LLM serving. Consider MoE load balancing (Section 3.2, Example 1). DeepSeek's ecosystem illustrates the full scientific process:
>
> - **EPLB** is an optimization-informed heuristic using workload statistics to reorder and replicate experts. It solves no optimization at runtime, yet its design is guided by the assignment problem structure. EPLB is deployed in DeepSeek V3/R1 production---described as ``battle-tested in production.''
>
> - **LPLB** directly solves an LP per batch to redistribute tokens optimally. Its embedded GPU solver achieves ${\sim}100\mu s$ per solve for intra-node optimization---small relative to decode steps (${\sim}30$--$100ms$). LPLB is in the early research stage, demonstrating that direct LP solving fits within the time budget.
>
> This progression---formulation $\rightarrow$ optimization-informed heuristic (EPLB, deployed) $\rightarrow$ direct optimization (LPLB, research)---is the scientific process we advocate. The LP formulation made explicit \emph{what} to optimize (minimize max per-GPU load) and what structure to exploit (redundancy topology), guiding practical heuristic design.
>
> The same pattern applies to DP load balancing (Example 2): the integer programming formulation reveals that short-horizon forecasts suffice for near-optimal balancing, directly yielding practical algorithms without full output length prediction.
>
> To directly answer the reviewer's deployment criterion question, we propose the following guideline for the revision:
>
> - **Decision frequency vs solver latency**: When the solver runs faster than the decision interval (e.g., $100\mu s$ LP vs.\ $30ms$ decode step), direct deployment is viable.
> - **Structural insights for heuristic design**: When direct solving is too slow, the formulation still reveals binding constraints and solution structure---enabling fast approximations with guarantees (as in bid-price policies from Online LP, or EPLB's optimization-guided design).
> - **Gap to theoretical bound**: When the gap between heuristic performance and the theoretical optimum is large ($>$20\%), principled approaches are justified; when small ($<$5\%), heuristics are near-optimal and effort is better directed elsewhere.
>
> We will incorporate this discussion into the revision.

---

> > ### Author Rebuttal · Reviewer_xynm · 2026-04-05
> >
> > Thank you authors for the rebuttal. I think my questions have been addressed.

---

### Decision · Program_Chairs · 2026-04-30

**Decision:**

Accept (regular)

**Comment:**

This is a constructive and timely position paper that effectively bridges ML systems engineering with Operations Research, making it a perfect fit for the ICML Position Paper track.

Initially, reviewers raised valid practical concerns regarding solver overhead, problem tractability, and system adaptability under dynamic workloads. The authors provided a rebuttal that systematically resolved all concerns, leading to reviewer support and score increases.

Specifically, the authors clarified the computational complexity of each serving problem to prove real-time viability, demonstrating that some models (like MoE load balancing) can be solved in microseconds. Furthermore, they committed to enhancing the final revision with concrete deployment guidelines for practitioners and historical OR success stories (such as airline revenue management) to ground their methodology. The paper convincingly argues that formalizing LLM serving problems is a necessary evolution for the field and will undoubtedly stimulate valuable cross-disciplinary discussion.